# Surgical Anatomy and Dissection of the Hypogastric Plexus in Nerve-Sparing Radical Hysterectomy

**DOI:** 10.3390/diagnostics14010083

**Published:** 2023-12-29

**Authors:** Stoyan Kostov, Yavor Kornovski, Angel Yordanov, Rafał Watrowski, Stanislav Slavchev, Yonka Ivanova, Tosho Ganev, Hakan Yalçın, Ilker Selçuk

**Affiliations:** 1Research Institute, Medical University Pleven, 5800 Pleven, Bulgaria; drstoqn.kostov@gmail.com; 2Department of Gynecology, Hospital “Saint Anna”, Medical University—“Prof. Dr. Paraskev Stoyanov”, 9002 Varna, Bulgaria; ykornovski@abv.bg (Y.K.); st_slavchev@abv.bg (S.S.); yonka.ivanova@abv.bg (Y.I.); 3Department of Gynecologic Oncology, Medical University Pleven, 5800 Pleven, Bulgaria; 4Department of Obstetrics and Gynecology, Helios Hospital Müllheim, 79379 Müllheim, Germany; rafal.watrowski@gmx.at; 5Faculty Associate, Medical Center—University of Freiburg, 79106 Freiburg, Germany; 6Department of Urology, Hospital “Saint Anna”, Medical University—“Prof. Dr. Paraskev Stoyanov”, 9002 Varna, Bulgaria; dr_ganev@yahoo.com; 7Department of Gynecologic Oncology, Ankara Bilkent City Hospital, Maternity Hospital, 06800 Ankara, Turkey; drhyalcin@yahoo.com (H.Y.); ilkerselcukmd@hotmail.com (I.S.)

**Keywords:** superior hypogastric plexus, inferior hypogastric plexus, hypogastric nerves, bladder nerve branches, pelvic splanchnic nerves, nerve-sparing radical hysterectomy

## Abstract

Radical hysterectomy is a central surgical procedure in gynecological oncology. A nerve-sparing approach is essential to minimize complications from iatrogenic injury to the pelvic nerves, resulting in postoperative urinary, anorectal, and sexual dysfunction. The hypogastric plexus (HP), a complex network of sympathetic and parasympathetic nerves, plays a critical role in pelvic autonomic innervation. This article offers a comprehensive overview of the surgical anatomy of the HP and provides a step-by-step description of HP dissection, with a particular emphasis on preserving the bladder nerve branches of the inferior HP. A thorough understanding and mastery of the anatomical and surgical nuances of HP dissection are crucial for optimizing outcomes in nerve-sparing gynecologic-oncological procedures.

## 1. Introduction

The pelvic autonomic nerve system is a complex network divided into three parts—the sympathetic, parasympathetic, and enteric nervous systems. The enteric nervous system controls the functions of the gastrointestinal tract and is not associated with the autonomic innervation of the pelvis [1]. Therefore, the sympathetic and parasympathetic nerve plexuses innervate the pelvic viscera [2]. The superior hypogastric plexus (SHP), sympathetic trunk, hypogastric nerves (HNs), and most parts of the inferior hypogastric plexus (IHP) contribute to the sympathetic system of the pelvis. In contrast, the pelvic splanchnic nerves (PSNs) (arising from the S2–S4 anterior sacral roots) provide the parasympathetic innervation of the pelvis. The IHP is an intricate plexus, which is formed by the HNs (mainly sympathetic and secondarily parasympathetic innervation), pelvic splanchnic nerves (parasympathetic), and sacral splanchnic nerves (sympathetic innervation from multiple tiny nerve fibers originating from the sacral sympathetic ganglions of the sympathetic trunk) [3,4,5,6,7]. Gynecologic oncologic procedures such as paraaortic (inframesenteric) or presacral lymphadenectomy and radical hysterectomy are often associated with an injury to the sympathetic and parasympathetic branches of the autonomic nervous system [6,7]. Injury to the pelvic autonomic nervous system leads to long-term postoperative complications such as urinary, anorectal, or sexual dysfunction (e.g., reduced vaginal lubrication and libido) [6,8]. Therefore, nerve-sparing procedures have gained increasing recognition in gyneco-oncological surgery [7,9,10,11,12]. Studies have reported that nerve-sparing radical hysterectomy (type C1) is associated with a similar oncological outcome compared to conventional radical hysterectomy (type C2). Moreover, the nerve-sparing procedure is associated with minimized surgical-related pelvic dysfunction [13,14]. Despite recent surgical developments in nerve-sparing radical pelvic surgery (especially radical hysterectomy), the rate of postoperative complications and quality of life remain unsatisfactory [12]. The reason could be the complex anatomy of the IHP and difficulties in the dissection and preservation of the IHP with its branches to the target organs. The aim of the present article is to demonstrate the dissection of the SHP and IHP in nerve-sparing radical hysterectomy type C1 [15]. The anatomy, development of avascular spaces, and surgical technique are highlighted in detail.

### 1.1. Anatomical Nomenclature for the Parametrium

Terms such as “superior” versus “inferior”, “anterior” versus “posterior”, and “medial” versus “lateral” will be used. The terms superior and inferior define the anatomical aspect in the surgical supine position.

The term “parametrium”, which refers to the fatty lymphoid tissue around the uterus (including the uterine body and cervix), defines three parametria—lateral, ventral, and dorsal parametria [15].

The lateral parametrium consists of the parauterine tissue (above the ureter) and paracervix tissue (below the ureter). The parauterine tissue, also known as the “uterine pedicle”, contains the uterine artery and the superficial uterine vein with the fatty lymphoid tissue. The paracervix (below the ureter) is separated by the vaginal vein (a synonym of the deep uterine vein) into a cranial part (pars vasculosa) and a caudal part (pars nervosa). The pars vasculosa may contain the branches of the internal iliac system (vaginal artery and inferior vesical artery) in addition to the vaginal or uterine vein. In contrast, the pars nervosa contains pelvic splanchnic nerves and the posteroinferior part of the IHP [7,15].

The dorsal parametrium consists of the uterosacral ligament; its superior (cranial) part is called the rectouterine ligament, and the inferior (caudal) part is called the rectovaginal ligament [4,11,15].

The ventral parametrium includes the vesicouterine ligament (VUL), which is medial and superior (anterior) to the distal ureter, and the vesicovaginal ligament (VVL), which is lateral and inferior (posterior) to the distal ureter. The term “vesicouterine ligament posterior leaf” can also be used as a synonym for the VVL [15].

The mesoureter is a mesentery-like structure containing fatty tissue, the HN, and the posterior part of the IHP. The mesoureter is located inferior (posterior in classical anatomical position) to the ureter and defines the lateral part of the dorsal parametrium [15]. The term “ureterohypogastric fascia” can also be used as a synonym for the mesoureter [11].

The paravaginal veins contribute to the structure of the VVL. They have been termed “inferior vesical veins” or “middle and inferior vesical veins” by different authors [4]. We prefer the term “paravaginal veins”. The vaginal vein (VV) is equal to the deep uterine vein, which drains the vaginal and vesical venous plexuses as a single tributary [4,9,11,15].

The paracolpium is described as a distinct fatty lymphoid anatomical structure adjacent to the vagina and inferior to the distal ureter. The VVL lies at the superolateral part of the paracolpium, close to the bladder, and the VVL is intertwined with the paracolpium. The VVL-paracolpium border contains the paravaginal veins and the bladder nerve branches (BNBs) of the IHP. The paravaginal veins lie parallel to the paracolpium and drains into the vaginal vein [4,5,15].

### 1.2. Anatomy of Pelvic Autonomic Nerve System (Hypogastric Plexus)

#### 1.2.1. Superior Hypogastric Plexus

The SHP is a retroperitoneal structure, formed caudal to the level of the inferior mesenteric artery origin and located slightly to the left of the midline. The SHP lies anterior to the aortic bifurcation and the left common iliac vein at the level of the sacral promontory/fifth lumbar vertebra body, between the left and right common iliac arteries [2,5]. Generally, the lower part of the SHP is located posterior to the sigmoid mesocolon at the right part of the superior rectal artery [5]. In 60% of cases, the SHP terminates at the sacral promontory, whereas in 40% of patients, it ends above the distal portion of the S1 vertebra body [6]. Variations in the morphology (condensation of nerves and shape) of the SHP are often encountered: a plexiform structure, two distinct nerves, a thick single nerve, a broadened band-like nerve trunk, etc. [1,2,16,17]. A dissection of the SHP from the posterior parietal peritoneum and loose connective tissue is usually straightforward [5]. The SHP is derived from two lateral roots and one median root. The median root of the SHP is the direct caudal continuation of the preaortic plexus (superior, intermesenteric, and inferior mesenteric plexuses) [2]. The lateral roots of the SHP are sympathetic and parasympathetic. The sympathetic root represents the lumbar splanchnic nerves, L3-4, which run through the lumbar ganglia of the sympathetic trunk on both sides. Interestingly, the parasympathetic pelvic splanchnic nerves (PSNs) also contribute to the formation of the SHP. These nerves have a cranial direction from the IHP via both the left and right hypogastric nerves [5]. In addition, the median root of the SHP, as the caudal continuation of the preaortic plexus, also contains sympathetic and parasympathetic fibers [2,5,16,17].

#### 1.2.2. Hypogastric Nerves

The hypogastric nerves (HNs) are paired nerves representing the direct caudolateral continuation of the SHP. Their origin (at the level of or above the sacral promontory) and morphology (either a distinct bundle of nerve fibers or a network of multiple thin nerve fibers) vary among the human population [1,6,7]. After originating from the SHP, the HNs run anterolateral to the presacral fascia and medial to the internal iliac vascular system. The HNs attach to the presacral fascia through the lateral ligaments of the rectum. After descending into the pelvis, the HNs pass approximately 5–20 mm infero-medially to the ureter and laterally to the rectovaginal ligament to enter the IHP [1,6,7,8]. The presence of an accessory hypogastric nerve is also possible [6]. The HN lies as a part of the mesoureter/ureterohypogastric fascia lateral to the dorsal parametrium (uterosacral ligament) [1,7,8]. Some authors suggested that the HN contains only sympathetic fibers [1]. However, the HNs contain mainly sympathetic fibers (descending from the SHP), but also a few parasympathetic fibers (ascending nerve branches from the pelvic splanchnic nerves and IHP) [1,5,8].

The inferior mesenteric plexus, superior hypogastric plexus, and hypogastric nerves are shown in Figure 1, Figure 2, Figure 3 and Figure 4.

#### 1.2.3. Pelvic Splanchnic Nerves

The PSNs arise from the anterior rami of the sacral roots 2-3-4 (S2–S4) and supply parasympathetic innervation to the pelvic structures via the IHP. As mentioned above, some PSN fibers have an ascending course and pass directly into the HNs and SHP [5,6,7]. Most PSNs have a caudomedial course from the pelvic sidewall at the posterior aspect of the internal iliac vein and cross the pelvic floor to enter into the IHP (3–4 cm lateral and 2–4 cm inferior to the Douglas pouch). In some cases, there is a strict conjunction between the middle rectal vessels and the PSNs [5,6,7]. However, the middle rectal artery is not always present, or it may share the same root with the vaginal and/or inferior vesical artery. The PSNs lie medial to the vesical vessels and run inferior to the vaginal vein (deep uterine vein) to join the IHP at the deep part of the lateral parametrium, also known as the paracervix [4,5,8]. The pathway of the bladder nerve branches of the IHP is thoroughly described during dissection.

#### 1.2.4. Inferior Hypogastric Plexus

The IHP is also known as the pelvic plexus or pelvic ganglion. It is a bilateral network of nerves, which lies in a sagittal plane slightly anterolateral to the mesorectum, lateral to the vagina and rectum, and posterolateral to the base of the bladder. The IHP can be dissected close to the parietal pelvic fascia [1,3,5,6,7]. The superior part of the IHP is located just below the lateral margin of the rectouterine/rectovaginal ligament within the leaves of the broad ligament retroperitoneally. The inferior part of the plexus is more condensed compared to the superior part [1]. The length (15–40 mm) and thickness (10–30 mm) of the IHP vary in medical literature [1,3,6,7,18,19,20,21]. The IHP can be found along the posterior aspect of the internal iliac artery and vein. Visceral branches related to the internal iliac artery and vein cross the plexus without a direct connection [6,20]. The shape of the plexus has been described as either triangular, quadrangular, or cross-shaped [1,19,20,21]. Mauroy et al. characterized the IHP as a triangle-shaped structure with a posterior base and an antero-inferior top, as well as three edges and angles of the IHP. However, this description seems somewhat speculative [20]. Traditionally, the anatomical description of the IHP has delineated two main plexuses, which are the anterior, comprising the vesical and uterovaginal plexuses, and the postero-medial, rectal plexus [5,6]. However, other authors categorize the IHP into three distinct plexuses: anterior (vesical), intermediate (uterovaginal), and posterior (rectal) [3]. Fuji et al. offered a different view, describing the IHP as being formed by a cross-shaped nerve bundle, with the lateral part being the PSN (S2–S4), medial (uterovaginal plexus), posterior (hypogastric nerve), and anterior bladder nerve branches (BNBs) [21]. In gynecologic-oncological surgery, special attention is paid to the PSNs and the BNBs, since the uterovaginal plexus is transected during uterine removal. On the other hand, the rectal plexus can easily be preserved.

### 1.3. Neuroanatomy of the Pelvis

Recently, Marc Possover introduced a new specialty called “neuropelveology”. It includes the diagnosis and treatment of the pathologies and dysfunctions of the pelvic nerves [22]. Both the somatic and the autonomic nervous systems are involved in the pelvis. The autonomic nervous system, also known as vegetative, consists of the sympathetic and parasympathetic systems. The pelvic somatic nerves originate from the ventral roots of the lumbar and sacral spinal nerves. These nerves innervate the skeletal muscles such as the external urethral and anal sphincter, the lower abdominal wall, and the limbs. As mentioned above, the sympathetic systems in the pelvis are mainly composed of the sympathetic trunk and the hypogastric plexuses, whereas the PSNs represent the parasympathetic system [2,8,23]. The sympathetic and parasympathetic systems of the pelvis innervate the descending colon, anorectum, uterus/vagina, and the bladder. The sympathetic system prevents micturition, defecation, and the flow of menstrual blood [2,8,23]. Possover reported that the superior part of the IHP innervates the vagina, uterine cervix, and the uterus. Surgical injury of this particular part will lead to vaginal and cervical hypoesthesia and diminished vaginal lubrication. Additionally, the superior part (mainly represented by the HN) is accountable for cervical and vaginal pain sensation. The middle part of the IHP is responsible for the sensation of fullness in the urinary bladder and the rectum. The inferior part (represented by the PSNs) causes the contraction of the terminal rectum and the contraction of the detrusor of the urinary bladder [24]. The sympathetic system plays a major role in women’s physiological sexual arousal. Injury to the pelvic sympathetic nervous system leads to the loss of patients’ libidos and decreased vaginal lubrication [25]. The PSNs also contribute to the motility of the rectum and sexual function [8,23]. In the lower urinary tract, the sympathetic and parasympathetic nerves mediate the autonomic nervous system, while the somatic nervous system is represented by the innervation of the pudendal nerve. The α-adrenergic sympathetic nerves stimulate the smooth internal sphincter, whereas the pudendal nerve innervates the external urethral sphincter. The stimulation of the SHP and the HNs leads to a modest increase in bladder pressure, contraction of the internal sphincter, and the inhibition of the detrusor contraction, allowing the urinary bladder to fill [26]. The PSNs control the evacuation of the urinary bladder, as they play a role in the facilitation of voluntary micturition. The stimulation of the PSNs causes the contraction of the detrusor muscle through muscarinic receptors. The relaxation of the internal sphincter (which is a result of the cessation of the sympathetic pelvic system) happens just before detrusor contraction [8,24,26]. Injury to the pelvic autonomic nerves during a step-by-step dissection of the superior and inferior hypogastric plexuses is clearly described in the presented article.

### 1.4. Step-by-Step Dissection of the Superior and Inferior Hypogastric Plexuses for Nerve-Sparing Radical Hysterectomy

A transperitoneal dissection through open midline laparotomy is discussed.

Step 1. Dissection of the retroperitoneum and identification of the ureter

A horizontal incision is made in the lateral parietal peritoneum, which starts from the round ligament and continues cranially up to the level of the common iliac artery/pelvic brim. The incision is slightly lateral to the ovarian vessels and medial to the external iliac vessels. The dissection of the retroperitoneal loose connective tissue and identification of the ureter at the base of the broad ligament posterior leaf inferior to the ovarian vessels are carried out.

Step 2. Development of Okabayashi’s medial pararectal space and identification of hypogastric nerves

The ovarian vessels and the posterior leaf of the broad ligament are temporarily preserved. This allows for more precise access to the medial pararectal space and a clear separation of the mesoureter from the broad ligament posterior leaf. The lateral border of Okabayashi’s medial pararectal space is formed by the ureter, mesoureter, and HN, whereas the medial border is defined by the rectovaginal ligament. The dissection of the space continues until its inferior border—the pelvic floor (iliococcygeus muscle). The anterior border is the lateral parametrium (paracervix), and the posterior border is the sacrum [20]. The ureter, together with its mesoureter, is dissected and mobilized laterally, so the medial pararectal space is developed. The HN is identified approximately 2 cm posterior to the ureter within the fascia sheet of the mesoureter, also called the ureterohypogastric fascia [11]. The initial separation of the mesoureter should be carried out close to the rectovaginal ligament until the HN is identified. A dissection that is too lateral to the HN can injure the HN during its identification. The Okabayashi pararectal space is shown in Figure 5.

Step 3. Dissection of Latzko’s lateral pararectal space and identification of pelvic splanchnic nerves, with posterior and intermediate parts of the inferior hypogastric plexus

The limits of Latzko’s lateral pararectal space are defined as follows: medially by the ureter; laterally by the internal iliac artery; inferiorly by the pelvic floor (iliococcygeus muscle); anteriorly by the lateral parametrium (paracervix); and posteriorly by the sacrum. Latzko’s lateral pararectal space is depicted in Figure 5. A dissection of the fatty tissue posteroinferior to the internal iliac vein at the lateral border of Latzko’s lateral pararectal space will reveal the bundles of the PSNs, which run obliquely in the caudomedial direction at the base of the lateral part of the lateral parametrium [27,28]. An appropriate and additional dissection of both pararectal spaces and the superior part of the presacral space exposes the end portion of the SHP, the HN, the PSNs, as well as the rectal plexus and the postero-superior part of the uterovaginal plexus. Additionally, in some patients, the postero-superior part of the vesical plexus could also be identified. These parts of the IHP can easily be identifiable by following the HN plate in a caudal direction, as illustrated in Figure 6.

Step 4. Dissection of other pelvic avascular spaces

Dissecting the pelvic avascular spaces allows for precise access to the pelvic retroperitoneal structures. Furthermore, the development of these avascular spaces emphasizes the three parametria of the uterus. Therefore, the surgeon could envisage the position of the IHP and its branches (Figure 6).

The dissection includes the following spaces [27,28]:Lateral paravesical space: This is located between the caudal edge of the external iliac vessels (lateral) and the obliterated umbilical artery/umbilicovesical fascia (medial).Medial paravesical space: This is positioned between the obliterated umbilical artery/umbilicovesical fascia (lateral), the lateral aspect of the bladder, and the vesicouterine/vesicovaginal ligament (medial), with the lateral parametrium/paracervix (posterior) and superior pubic ramus (anterior).Rectovaginal space: This is bounded by the dorsal parametrium/uterosacral ligament (lateral) between the rectum and vagina.Vesicovaginal space: This is bounded by the ventral parametrium/vesicouterine ligament (lateral) between the bladder and vagina [27,28].

The dissection of these avascular spaces (pararectal, paravesical, vesicovaginal, and rectovaginal) reveals the three parametria of the uterus (see Figure 7), and all three parametria can be observed in a postoperative uterus specimen (see Figure 8).

Step 5. Dissection of the superior hypogastric plexus

The dissection may start by using the already-identified HN as a landmark or by transecting the posterior parietal peritoneum above the sacral promontory at the cranial level. The sigmoid mesocolon is retracted in the opposite direction (caudolaterally, aiming at the left side), and the parietal peritoneum is incised slightly lateral to the common iliac artery and 1–3 cm above the sacral promontory. If the dissection starts at the medial pararectal space, the fatty and areolar connective tissue is dissected in the supero-posterior direction by following the HN until reaching the infero-posterior aspect of the SHP, or the fatty lymphoid tissue at the posterior aspect of the posterior parietal peritoneum can be dissected directly between the bilateral common iliac arteries anterior to the left common iliac vein [8,21,29]. The inferior border of the presacral (retrorectal) space is the pelvic floor (anococcygeal ligament). The lateral border of the presacral space differs according to the dissection level, where at the most cranial (superior) level, the border is on the left–left common iliac vein, whereas the right border is at the right common iliac artery; nevertheless, the bilateral ureter lies at the lateral edges of the presacral space [28]. The SHP is located mainly between the common iliac arteries, anterior to the sacral promontory, and the left common iliac vein is at the caudal level of the origin of the inferior mesenteric artery [16,17,29]. A vessel loop is placed around the SHP. Following the inferior (caudal) limit of the SHP at the retrorectal plane, the paired HNs are easily identified (Figure 9). Surgeons should remain vigilant to prevent iatrogenic injuries to the ureters, SHP, HNs, inferior mesenteric artery, superior rectal artery, left common iliac vein, and middle sacral vessels during SHP dissection [16,17].

Injury of the SHP often occurs during blind paraaortic or presacral lymph node dissection, whereas the HNs are damaged during an en bloc resection of the dorsal parametrium (especially its caudal part—the rectovaginal ligament). Injuries to the SHP and the HNs can lead to reduced bladder compliance, obstipation, flatus, anorectal dysfunction, loss of patients’ libidos, and decreased vaginal lubrication [8,25,26].

Step 6. Dissection of the inferior hypogastric plexus

The plane of the HN or the ureterohypogastric fascia/mesoureter are used as landmarks to identify the postero-superior border of the IHP. The uterine pedicle, comprising the uterine artery and vein, can be either cut and ligated at the origin of the internal iliac artery or left intact. If it is cut, the uterine pedicle is retracted craniomedially, and the ureteral tunnel space is developed (medially with the uterine cervix and vaginal fornix and laterally with the superomedial aspect of the ureter) [21]. The ureter can be dissected entirely and mobilized from the VUL, which is located anteromedial (superomedial in surgical supine position) to the distal ureter. Thus far, the ureter has served as the primary landmark for identifying the autonomic nerve structures, which are situated posterior (inferior in surgical supine position) to the ureter. Nonetheless, after the ureter is separated from the VUL and subsequently retracted laterally, other landmarks become necessary to ascertain the anatomical positions of the pelvic autonomic nerve structures [30,31]. One such landmark is the vaginal vein, also known as the deep uterine vein. By transecting and slightly medially retracting the deep uterine vein, the underlying pelvic splanchnic nerve structure, which lies in the sheet of the paracervix in a caudomedial direction, can be easily identified. The pelvic splanchnic nerves, originating from the anterior branches of the second and third sacral roots, are identifiable just beneath the vaginal vein. These nerves converge with the HN to form the IHP [7,9,18,21,30].

The IHP is damaged during the resection of the paracervix, rectovaginal ligament, and the VVL. Moreover, the PSNs could be injured during systematic pelvic lymph node dissection. Damage to the sympathetic fibers of the IHP may cause a lack of bladder urine sensation and sexual disorders. Injury to the rectal plexus of the IHP leads to obstipation and anorectal dysfunction. Damage to the parasympathetic fibers (PSNs) will probably lead to a hypocontractile bladder with decreased sensation and urinary retention [8,24,26].

Step 7. Dissection of the spaces near the ventral parametrium (paravaginal space) and lateralization of the distal ureter

Preserving the BNBs, also termed “inferior hypogastric plexus vesical bundles”, is crucial but the most challenging part of the pelvic nerve-sparing procedure, demanding a meticulous dissection nearby the VUL and VVL. Yabuki’s fourth space, or the more caudally located Okabayashi’s paravaginal space, is developed between the ureter (laterally) and anterolateral part of the upper vagina (medially), which are covered superiorly by the VUL (Figure 10). Therefore, the Yabuki’s space is bounded by the VUL (medial) and the ureter (lateral). The Okabayashi paravaginal space is deeply located and bounded by the paracolpium (medial) and the VVL (lateral) [4,21,32,33]. The VUL is transected at the bladder margin (Figure 11), and the vessels of the ligament, including the superior vesical vein and cervicovesical vessels, are cut and ligated. The superior vesical vein is located above the ureter and drains into the superficial uterine vein [4,21,30]. The superior vesical artery is a good anatomical landmark to identify the bladder margin of the VUL. Subsequently, the ureter is completely dissected laterally from the lateral vaginal wall to reach the infraureteric paracervix and paracolpium, where the vesical branches of the IHP run posterolateral to the distal ureter.

Step 8. Dissection of the bladder nerve branches of the inferior hypogastric plexus at the paracolpium

After the complete lateral dissection of the distal ureter from the upper vagina, the VVL, the paravaginal veins (vesico-vaginal veins), and the paracolpium are identified. It is important to note that the BNBs of the IHP run posterolateral to the distal ureter and parallelly lateral to the paracolpium together with the paravaginal veins. The BNBs are also located between the VVL and the paracolpium [4,18,21,30]. The paravaginal veins serve as essential landmarks for identifying the BNBs. Developing the Okabayashi paravaginal space deeply between the VVL (lateral) and paracolpium (inferomedial) allows for an easier and safer dissection of the BNBs of the IHP [4] (Figure 11). The paravaginal veins, which are part of the VVL (vesicouterine ligament posterior leaf) and drain into the vaginal vein (deep uterine vein), are cut and ligated together with the ligament at the level of the bladder (Figure 11 and Figure 12). At this step, the BNBs are preserved while cutting the VVL. The uterovaginal plexus is transected between the IHP and the cervix/vagina by entering the Fujii’s space (Figure 12). This space is above the level of the HN, and it is bounded laterally by the BNBs and medially by the vagina [9,21,30,31]. If the BNBs are separated from the paracolpium and fully mobilized laterally, that allows access to the inferomedial fatty tissue of the paracolpium (Figure 13, Figure 14, Figure 15 and Figure 16). BNBs are often injured during the dissection and blind transection of the VVL. The nerves could be also transected during paracolpium dissection and the resection of the vagina. Damage to the BNBs will cause a hypocontractile bladder with decreased sensation, impaired voluntarily voiding, and urinary retention [8,24,26].

The final view of the nerves after uterine removal is shown in Figure 17.

## 2. Discussion

Many different radical approaches for the surgical treatment of the uterine cervix have been delineated since the introduction of the radical hysterectomy [9,32,34,35,36,37]. Clark was the first to introduce the concept of radical hysterectomy at Johns Hopkins Hospital in 1895 [34]. Thereafter, in 1912, Wertheim introduced the first systematized work of radical hysterectomy on 500 patients with cancer of the uterine cervix [35,37]. In 1921, at the Kyoto Imperial University of Japan, Okabayashi showed a different and detailed surgical concept of radical hysterectomy. Okabayashi’s radical hysterectomy is characterized by a wide resection of the parametrial tissue and a dissection of the posterior leaf of the VUL. Okabayashi also described two avascular spaces (pararectal and paravaginal), which are still in use [21,32]. Later, Yabuki et al. stated the fourth space [33]. Yabuki’s paravaginal avascular space is cranial to Okabayashi’s paravaginal space and provides direct access to the BNBs of the IHP and VVL after the craniomedial traction of the uterine pedicle during type C1 nerve-sparing radical hysterectomy.

Since the work of Wertheim in Europe, the procedure has undergone many modifications. Meigs included the bilateral systematic pelvic lymphadenectomy as an integral part of the radical hysterectomy [38]. Piver et al. subdivided radical (extended) hysterectomy into five classes based on the extent of the resection of the three parametria and the vagina [39]. This classification was used for many years until the introduction of the new classification of radical hysterectomy by Querleu and Morrow. The new classification divides radical hysterectomy into four types (A-D), which are further subdivided into subtypes. Type C1 represents nerve-sparing radical hysterectomy [40]. The importance of pelvic autonomic nerve preservation during extended hysterectomy was introduced in classification for the first time. However, in 1944, Okabayashi mentioned that sparing the pelvic autonomic nerves would be the future challenges of the procedure, and Japanese surgeons defined a nerve-sparing radical hysterectomy technique [21]. Conventional radical hysterectomy (type C2 according to Querleu and Morrow and type III-IV according to Piver-Rutledge) is associated with a high rate of long-term postoperative complications as a result of an injury to the pelvic autonomic nerves—bladder dysfunction and anorectal or sexual dysfunction. Therefore, the importance of nerve-sparing radical hysterectomy in selected patients, without compromising the oncological outcomes, has increased over the past few decades [41]. Many anatomical and surgical articles have attempted to identify and correct the pathways and anatomical positions of pelvic autonomic nerves [41,42,43,44,45]. The number of articles that illustrate and show the dissection of the IHP in gynecologic oncology surgery has also increased in recent decades. However, most European studies have shown the initial dissection of the HN and the postero-superior part of the IHP, which is followed by a further blind antero-inferior dissection without the clear identification of the PSNs and BNBs [46,47,48,49]. The precise and meticulous identification, mobilization, and transection of the two layers of the ventral parametrium with the separation of the BNBs is not always easily performed. Furthermore, in some cases, the dissection is carried out under additional magnification (x2.5) [21]. Therefore, most articles did not show vivid illustrations of the PSNs and BNBs [46,47,48,49]. Additionally, there are considerable differences and disparities among authors of the surgical anatomy of PSNs and IHP vesical fibers (vf).

However, Japanese authors successfully managed to exhibit the pathways and dissection of the PSNs and BNBs. The anatomy of the pelvic autonomic nerves was clearly illustrated either during cadaveric dissection or surgical procedures [11,18,21,33,42,50]. An article on nerve-sparing radical hysterectomy with intraoperative figures illustrating the pelvic autonomic nerves in detail was recently introduced in Europe [9]. Nevertheless, studies which delineate the surgical anatomy of the hypogastric plexus continue to be inappreciable in gynecological oncology surgery.

We believe that during nerve-sparing radical hysterectomy, the following statements should be mentioned:The development of the lateral avascular spaces from the bifurcation of the common iliac artery to the Okabayashi paravaginal space emphasizes the pelvic autonomic nerve system and pelvic vascular systems [51].

The lateral limit of the Okabayashi pararectal space is the ureter, together with the mesoureter and the hypogastric nerve, which lie on the same axis [27,28]. The medial limit of this space is a subject of disparity among surgeons. Many different medial limits of Okabayashi’s pararectal space have been described in the literature—the rectum, uterosacral ligament, rectovaginal ligament, the pouch of Douglas, or the lateral ligament of the rectum [27,52]. One might suggest that the medial limit of Okabayashi’s pararectal space should be the rectovaginal ligament, as the latter is the lateral limit of the rectovaginal space. As a result, the rectovaginal space and Okabayashi’s pararectal space will unite as one entity.

Some authors used the term “paravaginal space” to describe the area nearby the ventral parametrium [45]. The authors illustrated the VVL as a medial border and illustrated the ureter as a lateral border [45]. Querleu et al. also used the term “paravaginal space” and described its boundaries—the ureter together with the BNBs (lateral) and vagina (medial) [52]. It should be stressed that Okabayashi’s paravaginal space is not identical to Yabuki’s (Fourth) paravaginal space [27,41]. Yabuki’s space is located cranial to Okabayashi’s paravaginal space. Yabuki’s space provides the separation of the ureter from the VUL, and so Yabuki tried to identify the BNBs on the VVL [21,33,41]. The BNBs (IHP vesical fibers) are located parallel to the paracolpium on the lateral side of the vagina wall, and they lie at the border between the VVL and the paracolpium [4,27]. Some authors do not develop Okabayashi’s paravaginal space during nerve-sparing radical hysterectomy [18]. We believe that Okabayashi’s paravaginal space is an integral part of nerve-sparing radical hysterectomy, as it separates the VVL (vesicouterine ligament posterior leaf) from the paracolpium and BNBs more precisely [4,30]. Furthermore, the development of Fujii’s avascular space to transect the uterine branches of the IHP was mentioned in only two studies [21,31,41]. The separation and lateralization of the IHP vesical fibers (by transecting the uterovaginal plexus) from the uterus and vaginal wall is not feasible without entering this space [41].

2.The subdivision of the three parametria of the uterus preciously emphasizes the majority of surgical steps during nerve-sparing radical hysterectomy. Additionally, the pelvic autonomic nerve system is recognized in the caudal part of the three parametria [4,21,30,41]. The HN is located near the rectovaginale ligament, the IHP is identified near the deep part of the paracervix, and one of the branches of the PSNs is also located in the paracervix just below the vaginal vein. BNBs can be identified just medial to the VVL (Figure 18).

3.We prefer the term “paravaginal veins”, which defines the (para)vaginal and vesical veins. Paravaginal veins are considered as a paravaginal anastomotic venous plexus rather than as separate veins (middle or inferior vesical vein) [4,41]. As mentioned above, the paravaginal veins are part of the VVL in most cases (Figure 19 and Figure 20).

4.Precise knowledge of the anatomical position of the IHP, especially the BNBs, is needed (Figure 21). The BNBs leave the IHP in an anterior direction, caudolaterally, and pass between the VVL and the paracolpium. They run parallel to the paravaginal veins and paracolpium [4,41]. In addition, some BNBs may arise caudomedially from the IHP, but most of the vesical fibers run caudolaterally towards the bladder base. The BNBs become visible just after the complete division of the paravaginal veins and the VVL [4].5.It should also be stressed that there is a possible presence of the neurovascular bundle of Walsh (first described in males during nerve-sparing prostatectomy) among females. The bundle of Walsh originates from the caudal portion of the IHP, runs in the posterior lateral aspect of the Denonvillier fascia, and descends along the rectoprostatic septum. The bundle further passes towards the prostatic apex and the urethral sphincter and ends at the penile cavernous bodies [53,54,55,56,57]. In men, the preservation of the bundle dramatically decreases the incidence of erectile dysfunction [53,54,55,56,57]. Although it is mainly described during nerve-sparing prostatectomy, there are several gynecological studies which also mention it. Kim reported that, in females, the bundle passes anterior to the rectogenital fascia and then runs in the parametrium [58]. Another study reported that the bundle of Walsh is located posterolateral to the vagina [59]. However, more surgical and anatomical studies are needed in order to investigate the possible presence of the neurovascular bundle of Walsh among the female population.6.Using thermal energy (electric or ultrasonic energy) during the dissection of autonomic nerves in the pelvis increases the risk of thermal injury and nerve edema. Especially at the deep ventral parametrium (VVL), during infraureteric paravaginal dissection, using scissors and suture ligating the bleeding zones will improve the function of the BNBs of the IHP. Electric or ultrasonic energy during the dissection of the VVL will potentially lead to thermal trauma in the BNBs [60,61,62]. New methods and surgical tools have been introduced in order to avoid thermal nerve injury. Zhao et al. reported that an ultrasonic scalpel combined with a vascular clip in parametrial management is associated with improved postoperative bladder function during nerve-sparing radical hysterectomy [61]. Recently, tissue-selective dissection with a water jet was introduced in gynecologic oncology surgery. Meshkova et al. reported that water jet tissue dissection during nerve-sparing radical hysterectomy contributes to the most atraumatic dissection of the autonomic nervous system [62].

## 3. Conclusions

The dissection and preservation of the SHP and HNs are more accessible than identifying the IHP. The IHP interacts with all three uterine parametria (dorsal, lateral, and ventral) and most vessels deriving from the internal iliac system. Achieving a clear visualization of the uterovaginal and vesical nerve branches of the IHP, as well as the vessels in the ventral parametrium (the superior vesical vein, paravaginal veins), can be difficult and complex, especially in patients with a high body mass index. Therefore, surgeons are advised to rely on anatomical landmarks to ensure the accurate localization and preservation of neural structures, particularly the BNBs of the IHP. These landmarks include the ureter, the HN, the ureterohypogastric fascia/mesoureter, the middle rectal artery/vaginal artery (if it is present), the vaginal vein/deep uterine vein, and the paravaginal veins. By following these landmarks, surgeons can navigate the pelvic anatomy confidently, thereby reducing long-term morbidity, such as urinary or sexual dysfunction, after radical hysterectomy.

## Figures and Tables

**Figure 1 diagnostics-14-00083-f001:**
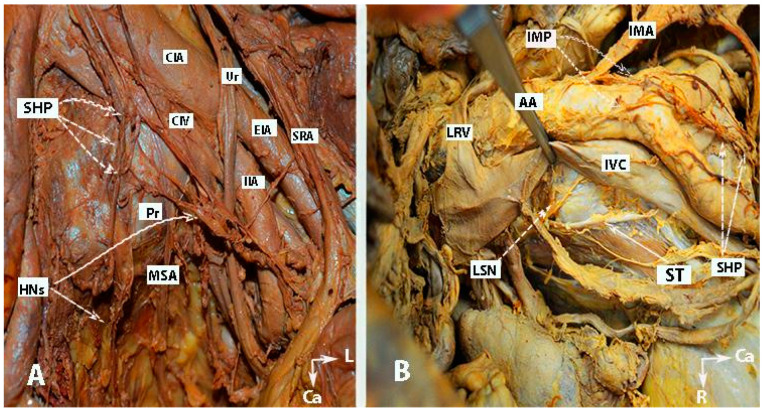
Inferior mesenteric plexus, superior hypogastric plexus, and hypogastric nerves (embalmed cadavers; authors’ own material). The superior hypogastric plexus has a plexiform structure on both figures. (**A**) Superior hypogastric plexus and hypogastric nerves are shown. (**B**) Lumbar splanchnic nerves (L3) and sympathetic trunk could be identified. SHP—superior hypogastric plexus; HNs—hypogastric nerves; Pr—promontory; CIA—common iliac artery; Ur—ureter; CIV—common iliac vein; EIA—external iliac artery; IIA—internal iliac artery; SRA—superior rectal artery; MSA—median sacral artery; IMP—inferior mesenteric plexus; IMA—inferior mesenteric artery; LRV—left renal vein; LSN—lumbar splanchnic nerve (L3); ST—sympathetic trunk; IVC—inferior vena cava; Ca—caudal; L—left.

**Figure 2 diagnostics-14-00083-f002:**
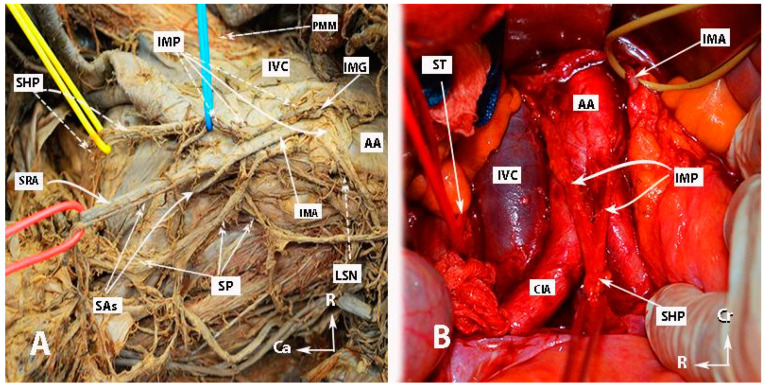
Inferior mesenteric plexus, superior hypogastric plexus, and hypogastric nerves (authors’ own material). (**A**) Embalmed cadaver; (**B**) open surgery. The superior hypogastric plexus has the morphology of a broadened band-like nerve trunk in both figures. SHP—superior hypogastric plexus; IMP—inferior mesenteric plexus; PMM—psoas major muscle; IVC—inferior vena cava; AA—abdominal aorta; IMG—inferior mesenteric ganglion; IMA—inferior mesenteric artery; LSN—lumbar splanchnic nerve (L3); SRA—superior rectal artery; SP—sigmoid plexus; Sas—sigmoid arteries; ST—sympathetic trunk marked with vessel loop; CIA—common iliac artery; Ca—caudal; R—right; Cr—cranial.

**Figure 3 diagnostics-14-00083-f003:**
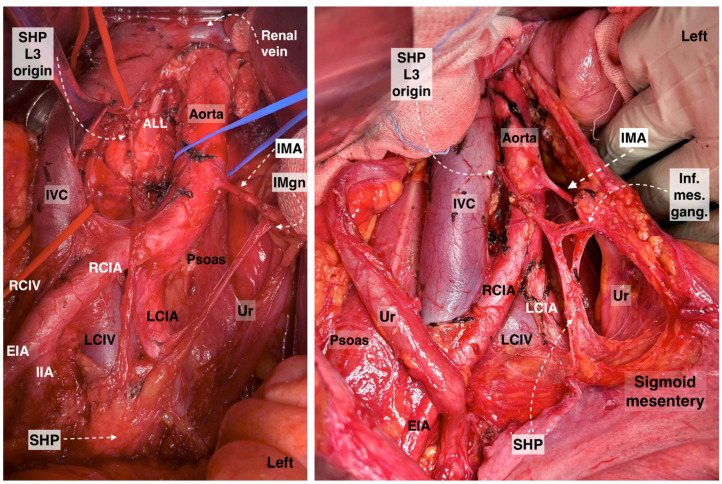
Superior hypogastric plexus, originating from the L3 paravertebral plexus and communicating nerve fibers with the inferior mesenteric ganglion. The caudal extension of the nerve fibers from the thoracic splanchnic nerves was excised during paraaortic lymph node dissection (surgical dissection by author IS) (L: lumbar; IVC: inferior vena cava; RCIA: right common iliac artery; RCIV: right common iliac vein; LCIA: left common iliac artery; LCIV: left common iliac vein; EIA: external iliac artery; IIA: internal iliac artery; IMA: inferior mesenteric artery; ALL: anterior longitudinal ligament; SHP: superior hypogastric plexus; Imgn: inferior mesenteric ganglion; Ur: ureter).

**Figure 4 diagnostics-14-00083-f004:**
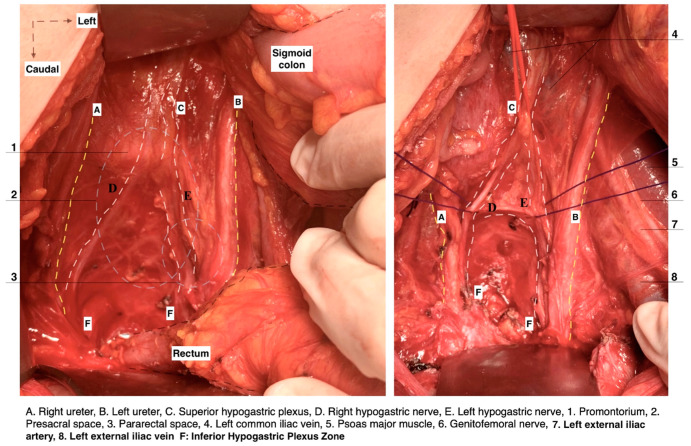
Superior hypogastric plexus lies anterior to the left common iliac vein and is divided into the right and left hypogastric nerves at the retrorectal plane. The hypogastric nerve runs approximately 2 cm posterior to the ureter (surgical dissection by author IS).

**Figure 5 diagnostics-14-00083-f005:**
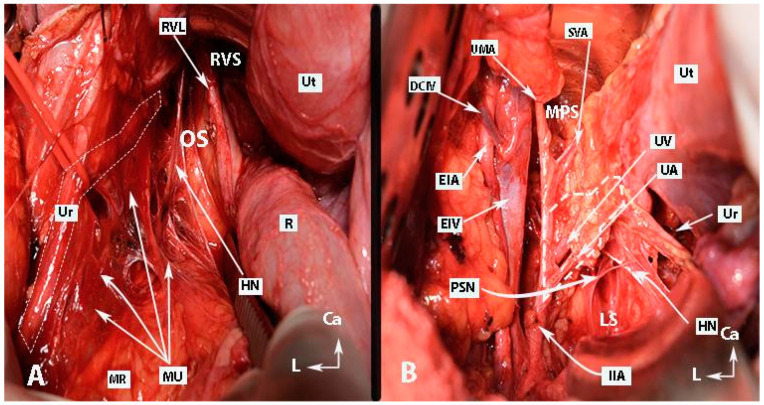
Pararectal spaces (authors’ own material—open surgery). (**A**) Okabayashi’s pararectal space. The dotted line shows the ureter. The ureter together with the mesoureter and hypogastric nerve are retracted laterally. (**B**) Latzko’s pararectal space. The ureter was retracted medially in order to identify one of the pelvic splanchnic nerves. The dotted shape shows the upper part of the lateral parametrium—the parauterine tissue—consisting of uterine artery/vein and lymphatic and fatty tissue. OS—Okabayashi’s pararectal space; RVS—rectovaginal space; Ut—uterus; R—rectum; RVL—rectovaginal ligament; HN—hypogastric nerve; MU—mesoureter; MR—mesorectum; Ur—ureter; EIA—external iliac artery; EIV—external iliac vein; PSN—pelvic splanchnic nerve; DCIV—deep circumflex iliac vein; UMA—umbilical artery; SVA—superior vesical artery; MPS—medial paravesical space; UV—uterine vein; UA—uterine artery; LS—Latzko’s space; IIA—internal iliac artery; CA—caudal; L—left.

**Figure 6 diagnostics-14-00083-f006:**
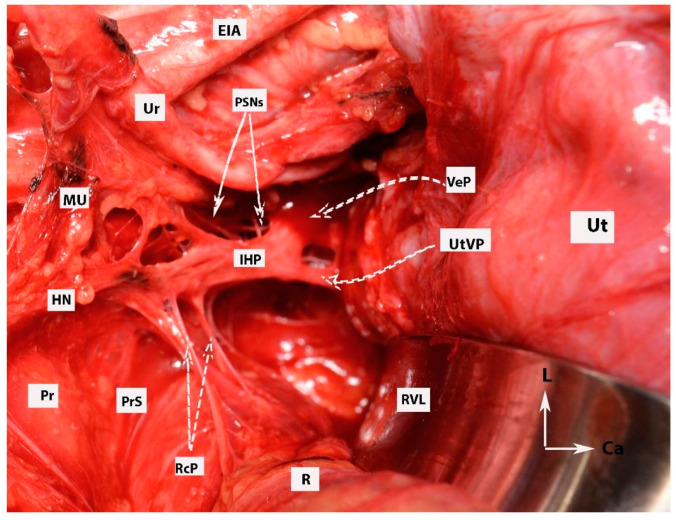
Dissection of posterior and postero-superior parts of the intermediate and anterior plexuses of the inferior hypogastric plexus (open surgery; authors’ own material). EIA—external iliac artery; Ur—ureter; MU—mesoureter; HN—hypogastric nerves; Pr—promontory; PrS—presacral space; Ut—uterus; R—rectum; IHP—inferior hypogastric plexus; PSNs—pelvic splanchnic nerves; VeP—postero-superior part of vesical plexus; UtVP—postero-superior part of uterovaginal plexus; RcP—rectal plexus; L—left; Ca—caudal.

**Figure 7 diagnostics-14-00083-f007:**
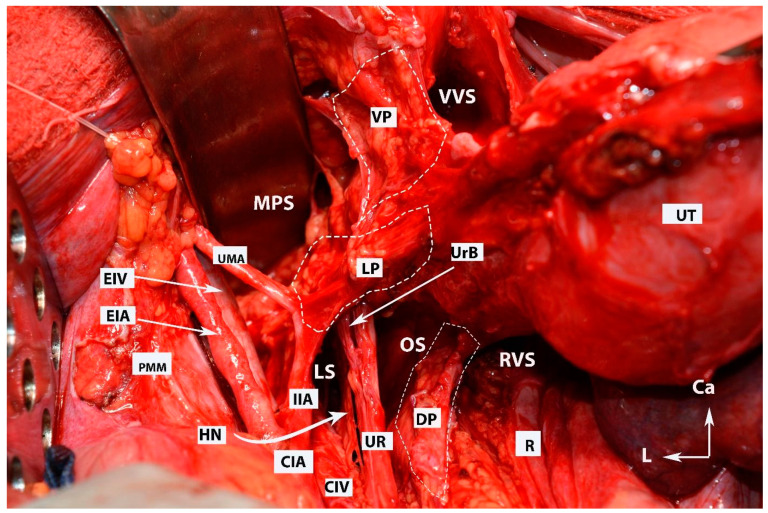
All three parametria after dissection of some of the female avascular spaces in the pelvis (open surgery; authors’ own material). VVS—vesicovaginal space; MPS—medial paravesical space; RVS—rectovaginal space; OS—Okabayashi’s pararectal space; LS—Latzko’s pararectal space; VP—ventral parametrium; LP—lateral parametrium; DP—dorsal parametrium; Ut—uterus; R—rectum; UMA—umbilical artery; EIV—external iliac vein; EIA—external iliac artery; PMM—psoas major muscle; IIA—internal iliac artery; HN—hypogastric nerve; Ur—ureter; UrB—ureteral branch of uterine artery; CIA—common iliac artery; CIV—common iliac vein; Ca—caudal; L—left.

**Figure 8 diagnostics-14-00083-f008:**
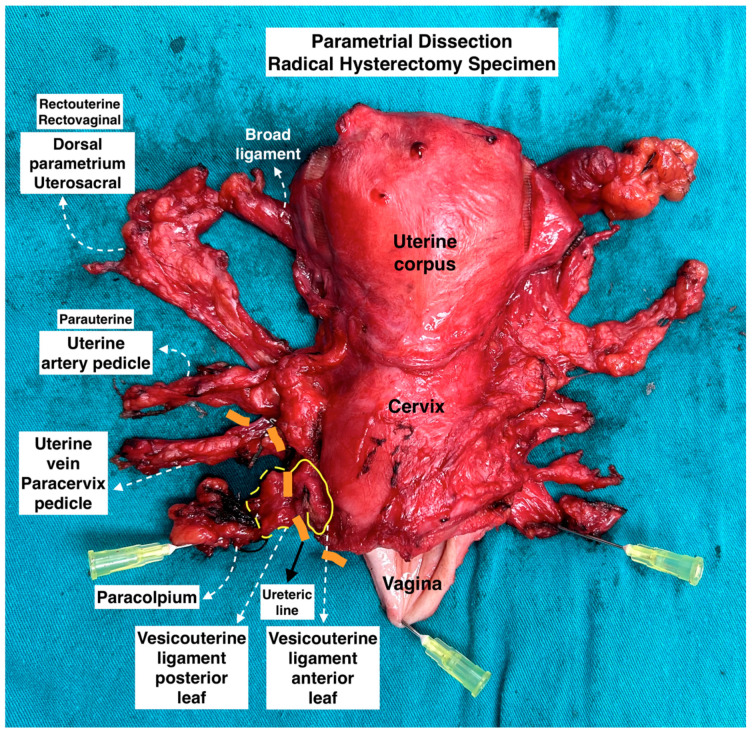
Parametrial dissection in radical hysterectomy specimen. (Surgical dissection by author IS.)

**Figure 9 diagnostics-14-00083-f009:**
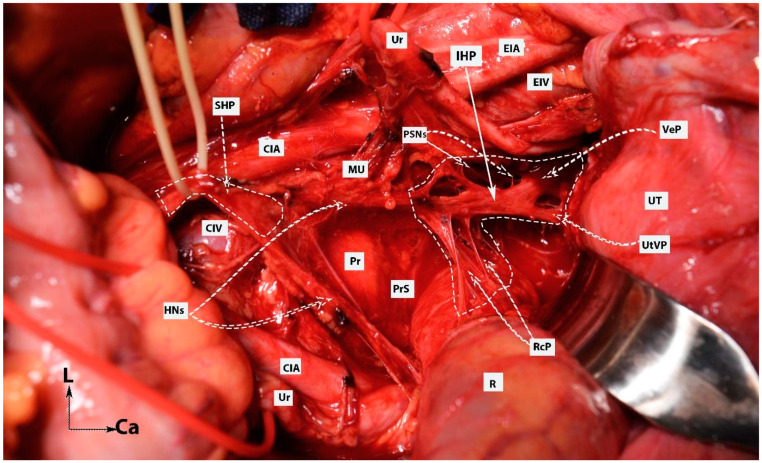
Dissection of superior hypogastric plexus, hypogastric nerves, and posterior, intermediate, and anterior parts of inferior hypogastric plexus (open surgery; authors’ own material). The already-dissected left hypogastric nerve is the key to encountering the inferior part of the superior hypogastric plexus. SHP—superior hypogastric plexus; IHP—inferior hypogastric plexus; CIV—common iliac vein; CIA—common iliac artery; HNs—left and right hypogastric nerves; Pr—promontory; PrS—presacral space; Ur—ureter; MU—mesoureter; PSN—pelvic splanchnic nerves; EIA—external iliac artery; EIV—external iliac vein; R—rectum; UT—uterus; VeP—postero-superior part of vesical plexus; UtVP—postero-superior part of uterovaginal plexus; RcP—rectal plexus.

**Figure 10 diagnostics-14-00083-f010:**
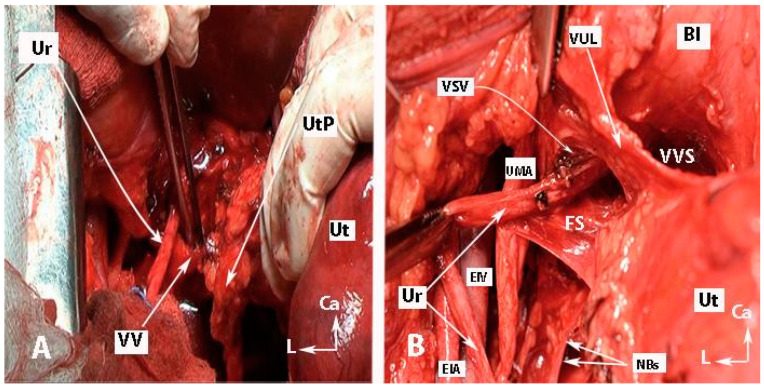
Transection of the vaginal vein and dissection of fourth space (open surgery; authors’ own material). (**A**) Identification of the vaginal vein. The uterine pedicle (uterine artery/vein together with the lymphatic and fatty tissue) was cut and mobilized medially. This facilitates entry into the ureteral tunnel and dissection of the vesicouterine ligament. (**B**) Development of the “Fourth space” between the ureter and the VUL. Vesical superior vein and vesicocervical vessels were coagulated and cut. Ur—ureter; UtP—uterine pedicle; VV—vaginal vein; EIV—external iliac vein; EIA—external iliac artery; UMA—umbilical artery; VUL—vesicouterine ligament; VVS—vesicovaginal space; VSV—vesical superficial vein (coagulated and cut); Bl—bladder; Ut—uterus; NBs—bladder nerve branches; FS—fourth space; CA—caudal; L—left.

**Figure 11 diagnostics-14-00083-f011:**
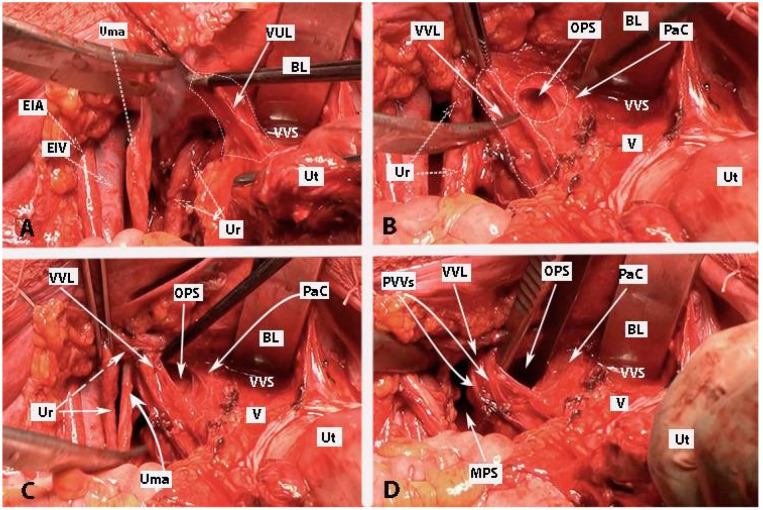
Development of Okabayashi’s paravaginal space (left body side—open surgery; authors’ own material). (**A**) The vesicouterine ligament is transected near the bladder. (**B**,**C**) Okabayashi’s paravaginal space is developed between the vesicovaginal ligament (lateral) and the paracolpium, the bladder nerve branches, and the paracolpial vein (medial). (**D**) The ureter was retracted laterally. The paravaginal veins, which are part of the vesicovaginal ligament, are identified, cut, and ligated. VUL—vesicouterine ligament; EIA—external iliac artery; EIV—external iliac vein; Ur—ureter; BL—bladder; Ut—uterus; Uma—umbilical artery; VVL—vesicovaginal ligament; OPS—Okabayashi’s paravaginal space; V—vagina; VVS—vesicovaginal space; PaC—paracolpium; PVVs—paravaginal veins; MPS—medial paravesical space.

**Figure 12 diagnostics-14-00083-f012:**
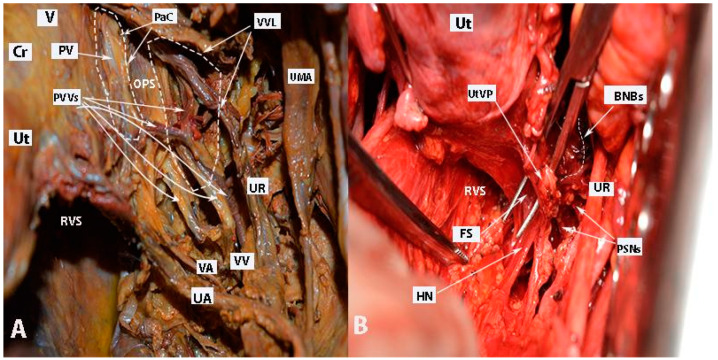
Paravaginal veins and Fujii’s space (right body side; authors’ own material). (**A**) Paravaginal veins, which represent paravaginal venous plexus with multiple veins draining as a single tributary into the vaginal vein. Paracolpial vein is also identified. It anastomoses with the paravaginal veins and drains into the vaginal vein (embalmed cadaver). (**B**) Fujii’s space is encountered in order to transect the uterovaginal plexus from the inferior hypogastric plexus. Using this maneuver, the remaining part of the IHP is easily lateralized (open surgery). Ut—uterus; Cr—cervix; V—vagina; UR—ureter; UA—uterine artery; VA—vaginal artery; VV—vaginal vein; PV—paracolpial vein; PVVs—paravaginal veins; VVL—vesicovaginal ligament; PaC—paracolpium; OPS—Okabayashi’s paravaginal space; UMA—umbilical artery; RVS—rectovaginal space; HN—hypogastric nerve; PSNs—pelvic splanchnic nerves; BNBs—bladder nerve branches; FS—Fujii’s space; UtVP—uterovaginal plexus.

**Figure 13 diagnostics-14-00083-f013:**
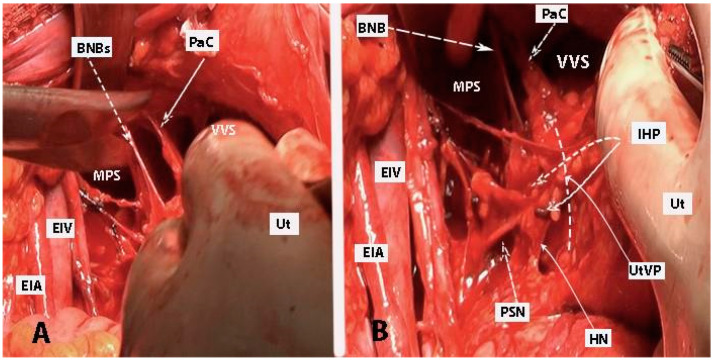
Bladder nerve branches are separated and fully mobilized from the paracolpium together with the paracolpial vein (left body side, open surgery; authors’ own material). (**A**) Dissection between the bladder nerve branches and the paracolpium. (**B**) IHP with the pelvic splanchnic nerve (S4), the BNBs, and the HN are identified. The dotted line shows the transection line of the uterovaginal plexus. BNBs—bladder nerve branches; EIA—external iliac artery; EIV—external iliac vein; Ut—uterus; MPS—medial paravesical space; PaC—paracolpium; VVS—vesicovaginal space; IHP—inferior hypogastric plexus; PSN—pelvic splanchnic nerve; UtVP—uterovaginal plexus.

**Figure 14 diagnostics-14-00083-f014:**
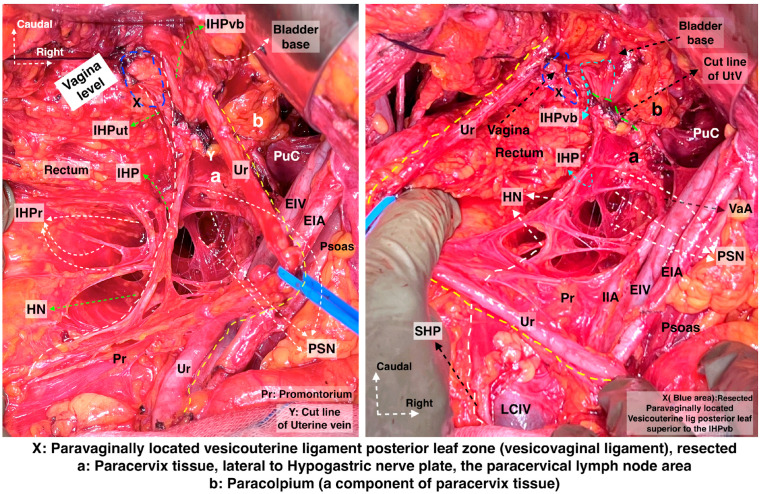
Hypogastric nerve running at the medial pararectal space and the pelvic splanchnic nerves, originating posterior to the internal iliac vessel system and running obliquely caudomedial from the lateral pararectal space. They unite to form the IHP lateral to the upper vagina and rectum. The IHP includes rectal, uterovaginal, and vesical nerve fibers. The vesical nerve fibers of the IHP run posterolateral to the distal ureter (surgical dissection by author IS) (Pr: Promontorium; HN: Hypogastric nerve; IHPr: Inferior hypogastric plexus rectal fibers; IHP: Inferior hypogastric plexus; IHPut: Inferior hypogastric plexus uterovaginal fibers; IHPvb: Inferior hypogastric plexus vesical bundles; PSN: Pelvic splanchnic nerves; Ur: Ureter; EIA: External iliac artery; EIV: External iliac vein; PuC: Pubococcygeus; IIA: Internal iliac artery; UtV: Uterine vein; VaA: Vaginal artery; LCIV: Left common iliac vein; SHP—Superior hypogastric plexus).

**Figure 15 diagnostics-14-00083-f015:**
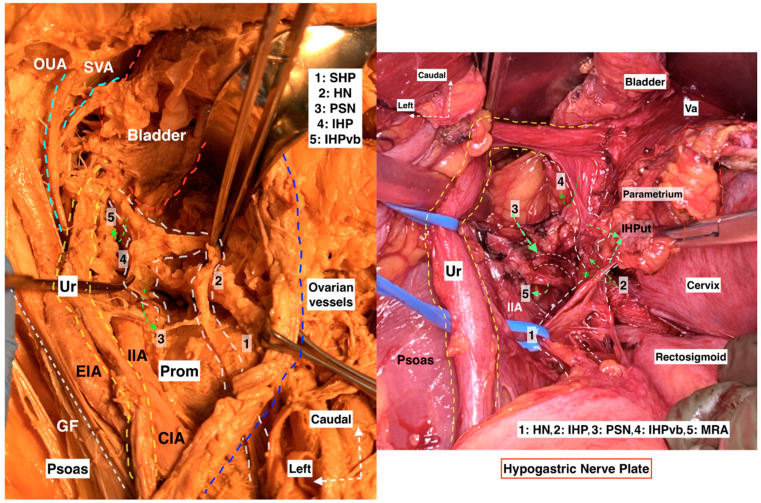
Hypogastric nerve plate. Dissecting the HN and following the HN plane facilitates the preservation of pelvic autonomic nerves ((**left**): cadaveric dissection; (**right**): surgical case) (cadaveric and surgical dissection by author IS) (CIA: Common iliac artery; EIA: External iliac artery; IIA: Internal iliac artery; Ur: Ureter; OUA: Obliterated umbilical artery; SVA: Superior vesical artery; GF: Genitofemoral nerve; Prom: Promontorium; Va: Vagina; IIA: Internal iliac artery; MRA: Middle rectal artery; SHP: Superior hypogastric plexus; HN: Hypogastric nerve; PSN: Pelvic splanchnic nerves; IHP: Inferior hypogastric plexus; IHPvb: Inferior hypogastric plexus vesical bundles; IHPut: Inferior hypogastric plexus uterine fibers).

**Figure 16 diagnostics-14-00083-f016:**
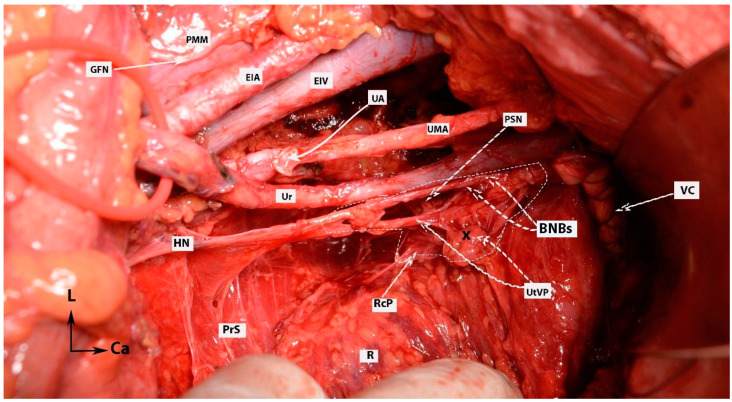
IHP with BNBs after nerve-sparing radical hysterectomy (authors’ own material from open surgery). PMM—psoas major muscle; EIA—external iliac artery; GFN—genitofemoral nerve; EIV—external iliac vein; UA—uterine artery (cut and ligated near its origin of the internal iliac artery); UMA—umbilical artery; Ur—ureter; PSN—pelvic splanchnic nerve; HN—hypogastric nerve; PrS—presacral space; R—rectum; RcP—rectal plexus; UtVP—uterovaginal plexus (x—most of the plexus is cut near the uterus/cervix and vagina); BNBs—bladder nerve branches; VC—vaginal cuff; L—left; Ca—caudal.

**Figure 17 diagnostics-14-00083-f017:**
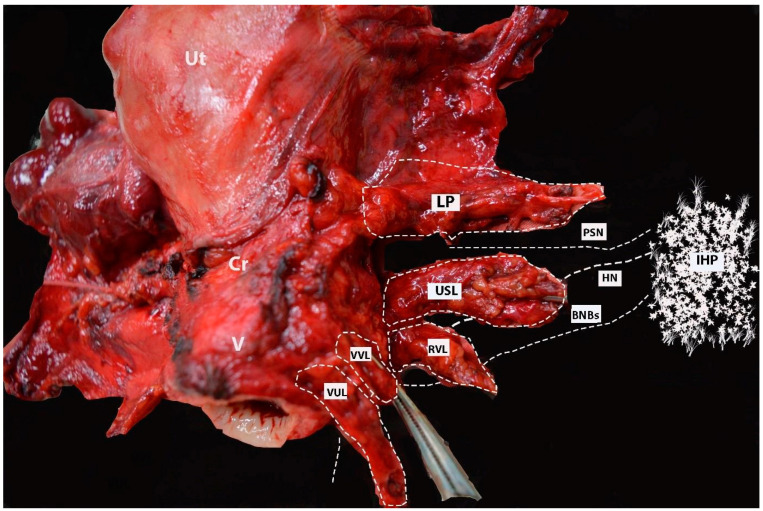
Postoperative specimen after nerve-sparing radical hysterectomy (authors’ own material). All three parametria (dorsal, lateral, and ventral) on postoperative specimen after nerve-sparing radical hysterectomy. The location and course of one of the pelvic splanchnic nerve, the HN, and the bladder nerve branches of the IHP are also illustrated. Lateral parametrium (the upper part—parauterine tissue—is removed; the lower part (paracervix) is removed just above the vaginal vein). The deeper part of the paracervix below the deep uterine vein is not resected. The uterosacral ligament and rectovaginal ligaments (dorsal parametrium) are transected. The vesicouterine and vesicovaginal ligament (ventral parametrium) are also transected. One of the pelvic splanchnic nerves passes just inferior to the vaginal vein. The HN passes close to the dorsal parametrium (underscoring the necessity of developing the Okabayashi pararectal space). The bladder nerve branches pass near the rectovaginal ligament and between the paracolpium (lateral to the paracolpium) and the vesicouterine and vesicovaginal ligaments (medial to both structures and below and parallel to the paracolpial vein). Ut—uterus; Cr—cervix; V—vagina; IHP—inferior hypogastric plexus; PSN—pelvic splanchnic nerve; BNBs—bladder nerve branches; HN—hypogastric nerve; LP—lateral parametrium; USL—uterosacral ligament; RVL—rectovaginal ligament; VVL—vesicovaginal ligament; VUL—vesicouterine ligament.

**Figure 18 diagnostics-14-00083-f018:**
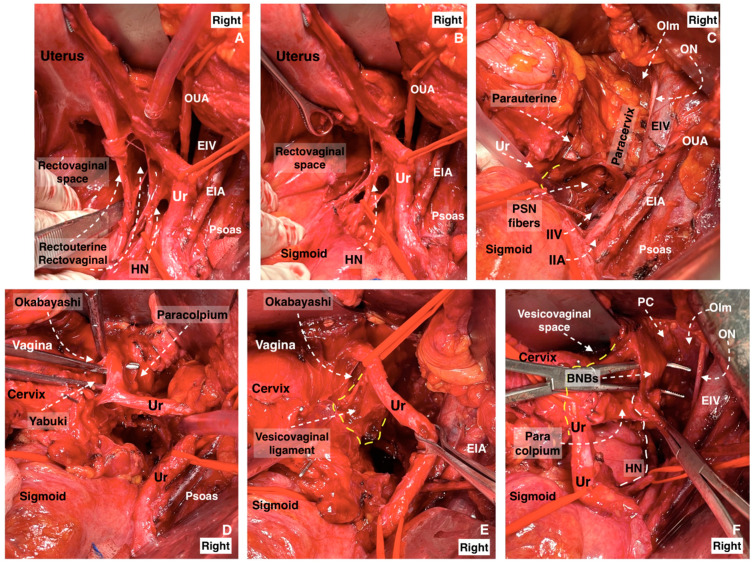
The dissection of the three parametria during nerve-sparing radical hysterectomy (surgical dissection by author IS). (**A**) Parametrial anatomy and hypogastric plexus. (**A**) Rectouterine and rectovaginal ligament with the hypogastric nerve, (**B**) excised rectouterine and rectovaginal ligament, also called the ureterohypogastric fascia sheet, (**C**) parauterine and paracervix tissue deep to the level of the pelvic floor, including the pelvic splanchnic nerve fibers, (**D**) Okabayashi and Yabuki paravaginal spaces after dissection of the vesicouterine ligament, (**E**) vesicovaginal ligament and Okabayashi paravaginal space, (**F**) dissection of the bladder nerve branches of inferior hypogastric plexus from the paracolpium (EIA: external iliac artery; EIV: external iliac vein; OUA: obliterated umbilical artery; Ur: ureter; HN: hypogastric nerve; IIA: internal iliac artery; IIV: internal iliac vein; ON: obturator nerve; OIm: obturator internus muscle; PSN: pelvic splanchnic nerves; BNBs: bladder nerve branches; PC: pubococcygeus).

**Figure 19 diagnostics-14-00083-f019:**
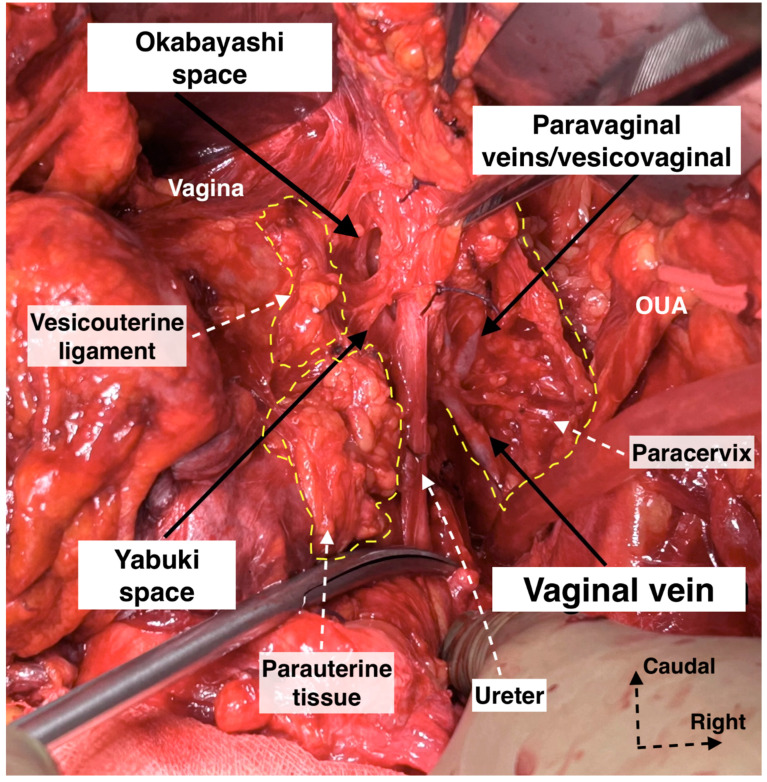
Supraureteric vesicouterine and parauterine tissue with Okabayashi and Yabuki paravaginal spaces and infraureteric paracervix (surgical dissection by author IS). Paravaginal (vesicovaginal) veins drain into the vaginal vein (deep uterine vein). OUA: obliterated umbilical artery.

**Figure 20 diagnostics-14-00083-f020:**
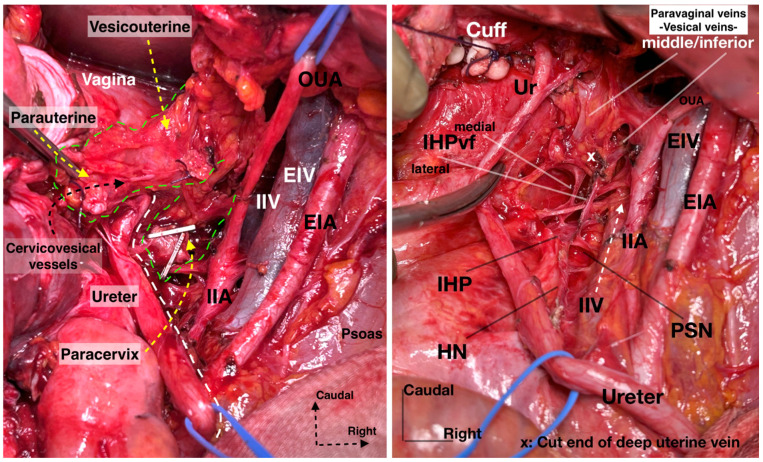
Supraureteric cervicovesical vessels and infraureteric vesicovaginal vessels with the medially and laterally extending vesical fibers of the inferior hypogastric plexus (surgical dissection by author IS). EIA: External iliac artery; EIV: External iliac vein; OUA: Obliterated umbilical artery; IIA: Internal iliac artery; IIV: Internal iliac vein; Ur: Ureter; HN: Hypogastric nerve; PSN: Pelvic splanchnic nerves; IHP: Inferior hypogastric plexus; vf: Vesical fibers.

**Figure 21 diagnostics-14-00083-f021:**
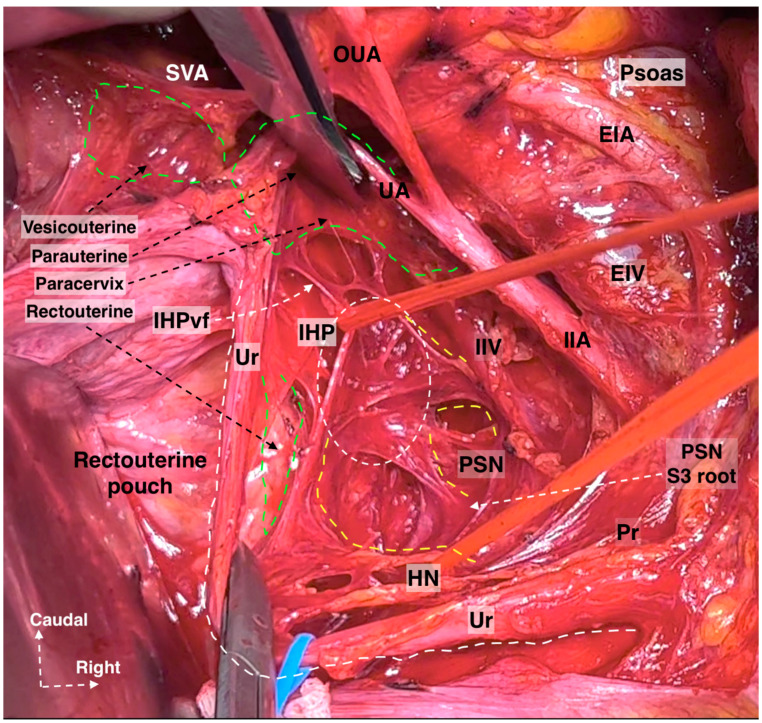
The formation of IHP by the HN and PSNs. The IHP is located at the deep medial part of the paracervix (surgical dissection by IS). EIA: External iliac artery; EIV: External iliac vein; OUA: Obliterated umbilical artery; UA: Uterine artery; SVA: Superior vesical artery; IIA: Internal iliac artery; IIV: Internal iliac vein; Ur: Ureter; HN: Hypogastric nerve; PSN: Pelvic splanchnic nerves; IHP: Inferior hypogastric plexus; vf: Vesical fibers; S: Sacral; Pr: Promontorium.

## Data Availability

The authors declare that all of the related data concerning the researchers are available.

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
