# Peer review of "Surgical Anatomy and Dissection of the Hypogastric Plexus in Nerve-Sparing Radical Hysterectomy"

_diagnostics, 2023, doi:10.3390/diagnostics14010083_

Round 1

Reviewer 1 Report

Comments and Suggestions for Authors

The anatomy description is too detail for an article of this nature. The authors should concentrate on the pelvic neuroanatomy relevant to the pelvic dissection during radical hysterectomy. The neurovascular bundle of Walsh was not described as it runs in the posterior lateral aspect of Denonvillier fascia and can easily be damaged during the anterior dissection. The description must be more clinical or functional and not just the insertion of an anatomical text.

Comments on the Quality of English Language

As above. The paper should be summarized with relevant anatomical details on the basis of autonomic dysfunction during radical hysterectomy.

Author Response

Dear Reviewer,
We are deeply grateful for your comprehensive review. We incorporated the recommended changes.
All incorporated changes are highlighted by using the Track and Changes in Word.

The anatomy description is too detail for an article of this nature. The authors should concentrate
on the pelvic neuroanatomy relevant to the pelvic dissection during radical hysterectomy.
Author’s Reply: The main purpose of the article is to show surgical anatomy and to describe step by
step the procedure. This is the reason, why pelvic anatomy is described in detail. However, we agree
with the reviewer that pelvic neuroanatomy (mainly functional and autonomic dysfunction after
nerve-sparing radical hysterectomy) should be at least mentioned, as this is the reason why we
perform nerve-sparing – better functional outcome and quality of life. Nevertheless, neuroanatomy
or neuropelveology also deals with pelvic nerves, the generation of pain, and putative sites of lesions
along the nerve pathway. This is not the main topic of the article. The aim is to show and illustrate
tips and tricks (development of avascular spaces; lines of resection) and surgical steps during NSRH.
The next chapter was inserted according to reviewer’s recommendations:
Recently, Marc Possover has introduced a new specialty called “neuropelveology” It includes the
diagnosis and treatment of pathologies and dysfunctions of the pelvic nerves [22]. Both the somatic
and the autonomic nervous systems are involved in the pelvis. The autonomic nervous system, also
known as vegetative, consists of the sympathetic and parasympathetic systems. The pelvic somatic
nerves originate from the ventral roots of the lumbar and sacral spinal nerves. These nerves
innervate the skeletal muscles such as the external urethral and anal sphincter, the lower abdominal
wall, and the limbs. As mentioned above the sympathetic systems in the pelvis is composed mainly of
the sympathetic trunk and the hypogastric plexuses, whereas the PSNs represent the
parasympathetic system [2, 8, 23]. The sympathetic and parasympathetic systems of the pelvis
innervates the descending colon, anorectum, uterus/vagina and the bladder. The sympathetic system
prevents micturition, defecation and the flow of menstrual blood [2, 8, 23]. Possover reported that
the superior part of the IHP innervates the vagina, uterine cervix and the uterus. Surgical injury of
this particular part will lead to vaginal and cervical hypoesthesia and diminished vaginal lubrication.
Additionally, the superior part (mainly represented by the HN) is accountable for the cervical and
vaginal pain sensation. The middle part of the IHP is responsible for the sensation of fullness of the
urinary bladder and the rectum. The inferior part (represented by the PSNs) causes contraction of the
terminal rectum and contraction of the detrusor of the urinary bladder [24]. The sympathetic system
plays a major role in women’s physiological sexual arousal. Injury to the pelvic sympathetic nervous
system leads to loss of patients’ libido and a decrease vaginal lubrication [25]. The PSNs also
contribute to the motility of the rectum and sexual function [8, 23]. In the lower urinary tract, the
sympathetic and parasympathetic nerves mediate the autonomic nervous system, while the somatic
nervous system is represented by the innervation of the pudendal nerve. The α-adrenergic
sympathetic nerves stimulate the smooth internal sphincter, whereas the pudendal nerve innervates
the external urethral sphincter. Stimulation of the SHP and the HNs leads to modest increase in
bladder pressure, contraction of the internal sphincter and inhibition of the detrusor contraction
allowing the urinary bladder to fill [26]. The PSNs control the evacuation of the urinary bladder, as
they play a role of the facilitation of voluntary micturition. Stimulation of the PSNs causes the
contraction of the detrusor muscle through muscarinic receptors. Relaxation of the internal sphincter

(it is a result of the cessation of the sympathetic pelvic system) happens just before detrusor
contraction [8, 24, 26]. Injury to the pelvic autonomic nerves during step-by-step dissection of the
superior and inferior hypogastric plexus is clearly described in the presented article.
The next references were inserted: 22, 23, 24, 25, 26.
Additionally, the next text was inserted during some of the surgical steps

Injury of the SHP often occurs during blind para-aortic or presacral lymph node dissection, whereas
the HNs are damaged during en-bloc resection of the dorsal parametrium (especially its caudal part-
the rectovaginal ligament). Injuries to the SHP and the HNs can lead to reduced bladder compliance,
obstipation, flatus, anorectal dysfunction, loss of patients’ libido and a decrease vaginal lubrication
[8, 25, 26].
Lines 376-381; page – 12

The IHP is damaged during resection of the paracervix, rectovaginal ligament and the VVL. Moreover,
the PSNs could be injured during systematic pelvic lymph node dissection. Damage to the
sympathetic fibers of the IHP may cause lack of bladder-urine sensation and sexual disorders. Injury
of the rectal plexus of the IHP leads to obstipation and anorectal dysfunction. Damage to the
parasympathetic fibers (PSNs) will probably lead to a hypocontractile bladder with decreased
sensation and urinary retention [8, 24, 26].
Lines 414- 419; Page- 13

BNBs are often injured during dissection and blind transection of the VVL. The nerves could be also
transected during paracolpium dissection and resection of the vagina. Damaged to the BNBs will
cause hypocontractile bladder with decreased sensation, impaired voluntarily voiding and urinary
retention [8, 24, 26].

Lines 478-480; Page - 15

The neurovascular bundle of Walsh was not described as it runs in the posterior lateral aspect of
Denonvillier fascia and can easily be damaged during the anterior dissection. The description must
be more clinical or functional and not just the insertion of an anatomical text.
As above. The paper should be summarized with relevant anatomical details on the basis of
autonomic dysfunction during radical hysterectomy.
Author’s Reply. The bundle of Walsh and Denonvillier fascia among women are debatable. The
Denonvillier fascia is found mainly in men, as most of the surgeons debate about its existence in
women. The neurovascular bundle of Walsh was also generally described in urology during nerve-
sparing prostatectomy. It’s existence in women is debatable. However, we agree with the reviewer
that it should be mentioned, as there are studies in gynecology (only a few), which also described the
bundle of Walsh. The next text was incorporated. We inserted it at the discussion part, as in the step-
by-step part, the explanation of the Bundle of Walsh will further confused readers.

It should be also stressed the possible presence of the neurovascular bundle of Walsh (first described
in males during nerve-sparing prostatectomy) among females. The bundle of Walsh originates from
the caudal portion of the IHP, runs in the posterior lateral aspect of the Denonvillier fascia and
descends along the rectoprostatic septum. The bundle further passes towards the prostatic apex and
the urethral sphincter and ends at the penile cavernous bodies [53-57]. In men, the preservation of
the bundle dramatically decreases the incidence of erectile dysfunction [53-57]. Although it is
mainly described during nerve-sparing prostatectomy, there are several gynecological studies, which
also mentioned it. Kim reported that in females the bundle passes anterior to the rectogenital fascia
and then runs in the parametrium [58]. Another study reported that the bundle of Walsh is located
posterolateral to the vagina [59].However, more surgical and anatomical studies are needed in order
to investigate the possible presence of the neurovascular bundle of Walsh among female population.

References from 53 to 59 were inserted.

We are grateful for your valuable time and effort in reviewing our manuscript.
Based on your useful and scientific comments, we believe our manuscript has been
improved to a higher level.

Reviewer 2 Report

Comments and Suggestions for Authors

Thanks to the authors for giving me the opportunity to review their manuscript. It is very well written, with beautiful iconography. Any surgeon who performs pelvic surgery of any kind should be familiar with this anatomy. I have a few comments:

- figure 6 DS is not present in the photo

- figure 17 SUL on the photo is USL (uterosacral ligament)

- figure 20 put in the text what IIV and IIA correspond to (internal iliac vein and internal iliac artery)

in the intro concerning functional and oncological results, I would add a meta-analysis reference such as 

Efficacy and oncologic safety of nerve-sparing radical hysterectomy for cervical cancer: a randomized controlled trial

Ju Won Roh, Dong Ock Lee, Dong Hoon Suh, Myong Cheol Lim, Sang Soo Seo, Jinsoo Chung, Sun Lee, Sang Yoon Park

J Gynecol Oncol. 2015 Apr;26(2):90-9

or

Efficacy of nerve-sparing radical hysterectomy vs. conventional radical hysterectomy in early-stage cervical cancer: A systematic review and meta-analysis.

Lee SH, Bae JW, Han M, Cho YJ, Park JW, Oh SR, Kim SJ, Choe SY, Yun JH, Lee Y.

Mol Clin Oncol. 2020 Feb;12(2):160-168.

Or another of your choice

In the discussion, I would like to draw the reader's attention (and in particular young surgeons) to the danger of coagulation and thermofusion instruments (Ligasure, etc.). They give off a great deal of heat, and it would be a pity to perform nerve sparing surgery and damage the nerve plexus by hyperthermia with conduction.

Comments on the Quality of English Language

minor corrections

Author Response

Dear Reviewer,
We are deeply grateful for your comprehensive review. We incorporated the recommended changes.
All incorporated changes are highlighted by using the Track and Changes in Word.

Thanks to the authors for giving me the opportunity to review their manuscript. It is very well
written, with beautiful iconography. Any surgeon who performs pelvic surgery of any kind should
be familiar with this anatomy. I have a few comments:
- figure 6 DS is not present in the photo
Author’s Reply: We agree with the reviewer. It is our mistake. Actually, the mistake is in figure 7. It
was corrected. It should be DP in the legend, not DS.

- figure 17 SUL on the photo is USL (uterosacral ligament)
Author’s Reply: It is our mistake! It should be like in the legend – USL. It was changed in the figure!

- figure 20 put in the text what IIV and IIA correspond to (internal iliac vein and internal iliac artery)
Author’s Reply: It was mentioned in the legends below figure 20.

in the intro concerning functional and oncological results, I would add a meta-analysis reference
such as 

Efficacy and oncologic safety of nerve-sparing radical hysterectomy for cervical cancer: a
randomized controlled trial
Ju Won Roh, Dong Ock Lee, Dong Hoon Suh, Myong Cheol Lim, Sang Soo Seo, Jinsoo Chung, Sun
Lee, Sang Yoon Park
J Gynecol Oncol. 2015 Apr;26(2):90-9

or
Efficacy of nerve-sparing radical hysterectomy vs. conventional radical hysterectomy in early-stage
cervical cancer: A systematic review and meta-analysis.
Lee SH, Bae JW, Han M, Cho YJ, Park JW, Oh SR, Kim SJ, Choe SY, Yun JH, Lee Y.
Mol Clin Oncol. 2020 Feb;12(2):160-168.
Or another of your choice

Author’s Reply: Both references of the recommended articles were incorporated in the
introduction. The next text was incorporated in the introduction

Studies reported that nerve-sparing radical hysterectomy (type C1) is associated with similar
oncological outcome compare to conventional radical hysterectomy (type C2). Moreover, nerve-
sparing procedure is associated with minimized surgical-related pelvic dysfunction [13, 14].
References number 13, 14 were incorporated.

In the discussion, I would like to draw the reader's attention (and in particular young surgeons) to
the danger of coagulation and thermofusion instruments (Ligasure, etc.). They give off a great deal
of heat, and it would be a pity to perform nerve-sparing surgery and damage the nerve plexus by
hyperthermia with conduction.
Author’s Reply: Thank you for the precious and important comment. We also think that it is very
important!
The next text was inserted:
Using thermal energy (electric or ultrasonic energy) during the dissection of autonomic nerves in the
pelvis increases the risk of thermal injury and nerve edema. Especially at the deep ventral
parametrium (VVL), during infraureteric paravaginal dissection using scissors and suture ligating the
bleeding zones will improve the function of the BNBs of the IHP. Electric or ultrasonic energy during
dissection of the VVL will potentially lead to thermal trauma of the BNBs [60-62]. New methods and
surgical tools in order to avoid thermal nerve injury have been introduced. Zhao et al. reported that
ultrasonic scalpel combined with vascular clip in parametrial management is associated with
improved postoperative bladder function during nerve-sparing radical hysterectomy [61]. Recently,
tissue-selective dissection with a water-jet was introduced in gynecologic oncology surgery.
Meshkova et al. reported that water jet tissue dissection during nerve-sparing radical hysterectomy
contributes to the most atraumatic dissection of the autonomic nervous sys-tem [62].
Lines: 703-114; Page- 25

References- 60, 61, 62 were inserted.

Comments on the Quality of English Language
minor corrections
Author’s Reply: We carefully revised the English grammar!

We are grateful for your valuable time and effort in reviewing our manuscript.
Based on your useful and scientific comments, we believe our manuscript has been
improved to a higher level.